# PeerJ

# Whale shark economics: a valuation of wildlife tourism in South Ari Atoll, Maldives

Edgar Fernando Cagua[1,2], Neal Collins[1,3], James Hancock[1] and Richard Rees[1]

[1] Maldives Whale Shark Research Programme, South Ari Atoll, Maldives
[2] Red Sea Research Center, King Abdullah University of Science and Technology, Thuwal, Saudi Arabia
[3] International Union for the Conservation of Nature, Gland, Switzerland

## ABSTRACT

Whale sharks attract large numbers of tourists, divers and snorkelers each year to South Ari Atoll in the Republic of Maldives. Yet without information regarding the use and economic extent of the attraction, it is difficult to prioritize conservation or implement effective management plans. We used empirical recreational data and generalized mixed statistical models to conduct the first economic valuation (with direct spend as the primary proxy) of whale shark tourism in Maldives. We estimated that direct expenditures for whale shark focused tourism in the South Ari Marine Protected Area for 2012 and 2013 accounted for US$7.6 and $9.4 million respectively. These expenditures are based on an estimate of 72,000–78,000 tourists who are involved in whale shark excursions annually. That substantial amount of income to resort owners and operators, and tourism businesses in a relatively small area highlights the need to implement regulations and management that safeguard the sustainability of the industry through ensuring guest satisfaction and whale shark conservation.

Corresponding author
Edgar Fernando Cagua,
edgar.cagua@kaust.edu.sa

## INTRODUCTION

In tropical locations around the world a new wildlife tourism industry has emerged in the last two decades that brings tourists in close proximity with whale sharks (*Rhincodon typus*; *Catlin et al., 2010b*). Due to the sharks' docile nature, patterns of seasonal aggregation (*Sequeira et al., 2013*), as well as accessibility, tourists are able to snorkel and scuba dive with unrestrained (or free-swimming) whale sharks. Whale sharks are listed as "Vulnerable" to extinction on the IUCN Red List of Threatened Species (*IUCN, 2014*); due to this, whale shark tourism has been hailed as an important income-generating alternative to consumptive or extractive uses of whale sharks such as shark finning or liver-oil processing (*Norman & Catlin, 2007*).

Tourism revenue can be considered a type of non-consumptive direct use value (*Catlin et al., 2013*; for a description of value types see *Turner et al., 2003*). The value of a natural location or a non-consumptive activity can be evaluated from a non-market perspective by

**Table 1 Previous economic valuation of whale shark tourism (in US million dollars).** Valuations reported in other currencies were converted to US$ using the average official rate for the year.

| Location (season duration) | Year | Total expenditure | Expenditure on WS excursions | Method | Reference |
|---|---|---|---|---|---|
| Belize (6 wks) | 2002 | $3.7 | – | Direct spend | *Graham (2003)* |
| Seychelles (14 wks) | 2003 | – | $1.2 | Contingent | *Cesar et al. (2004)* |
| | 2007 | $3.9–5.0 | – | Direct spend | H Newman et al., 2007, unpublished data[a] |
| Ningaloo (9 wks) | 1994 | $4.7 | $1.0 | Direct spend | *Davis et al. (1997)* |
| | 2004 | $13.3 | – | Unknown | *Norman (2005)* |
| | 2006 | $4.5 | $2.3 | Direct spend | *Catlin et al. (2010b)* |
| | 2006 | $1.8–3.5 | – | Substitution value | *Catlin et al. (2010b)* |

**Notes.**

[a] Cited in *Rowat & Engelhardt, 2007*.

using contingent (e.g., willingness to pay) and travel cost methods, or complimentary by using market-based valuations like those obtained by measuring expenditure. The *direct spend* method, which has been previously used to evaluate the impact of elasmobranch watching (*Anderson et al., 2011*; *Clua, Buray & Legendre, 2011*), provides a "minimal very conservative estimate of the economic value of natural areas" (*Wood & Glasson, 2005*). When data are available researchers use multipliers to also estimate the indirect effects in the economy (*Catlin et al., 2010b*). *Direct spend*, however, might also overestimate the value if it includes expenditures not exclusive of that resource. Therefore, tourism expenditure cannot be attributed to the natural resource if it is not the reason of the trip nor it influences the length of the stay. By estimating only the direct expenditure in whale shark excursions our valuation is closer to the substitution value, i.e., "the amount of expenditure that would be lost if whale shark tourism did not exist" (*Catlin et al., 2010b*).

The whale shark tourism industry was first organized at Ningaloo Reef in Western Australia in the late 1980s and early 1990s when operators began taking tourists mainly on diving excursions to swim with whale sharks when they appeared nearshore each year from May through June (*Colman, 1997*; *Davis et al., 1997*; *Catlin & Jones, 2010*). Whale shark tourism industries can now be found at numerous places worldwide—including the eastern Gulf of Mexico, Honduras, Belize, the Philippines, Mozambique, Seychelles, and the Maldives (*Quiros, 2005*; *Sequeira et al., 2013*). The burgeoning industry has made a strong economic case for conservation in that the sharks are worth more alive for tourism purposes than dead (*Cisneros-Montemayor et al., 2013*). However, the economics of whale shark tourism remains unclear apart from economic evaluations from Belize, the Seychelles and Ningaloo Reef (Table 1). Without this information it is difficult for localities with limited institutional powers—particularly in regards to environmental protection—to prioritize conservation of natural areas and implement effective management plans.

One popular location for whale shark tourism is the Republic of Maldives. Known for its abundance of sharks, rays, turtles, and cetaceans, the country is an iconic location

for marine wildlife tourism. While local populations historically used marine resources such as whale sharks for extractive purposes, the exact closure date of the Maldives whale shark fishery is unclear. *Sinan, Adam & Anderson (2011)* suggest that large shark fisheries for liver-oil extraction ceased in the 1960s, while *Anderson & Ahmed (1993)* reports it still happened in small-scale in the early 1990s. In 1993 the first valuation of the reef shark diving tourism industry was made public, and concerns about its vulnerability from pelagic fisheries precipitated a chain of legislation that ended with a national whale shark hunting ban in 1995 (Notice No: FA-A1/29/95/39) and the subsequent declaration of three Marine Protected Areas in 2009—Hanifaru Bay, Agafaru, and the South Ari Atoll Marine Protected Area (South Ari MPA).

The South Ari MPA is well-known regionally due to the occurrence of whale sharks throughout the year. Unlike the Hanifaru Bay MPA—one area in the Baa Atoll Biosphere Reserve with a management plan in place—the South Ari MPA's protected status is preliminary in that there is neither a management plan nor regulation in place yet. Anecdotal data suggest that tens of thousands of tourists participate in whale shark excursions there each year, however, no statistics exist that detail the extent of the industry or its economic benefit.

Without informed and effective management, wildlife tourism can have negative effects on wildlife like disruption of activity, injuring, and habitat alteration, ultimately damaging the resource it is intended to protect (*Green & Higginbottom, 2000*); as stakeholders overuse the resource, the long-term benefit is jeopardized (*Isaacs, 2000*; *Moore & Rodger, 2010*). Site-specific information and statistics are not only important to prioritize conservation, but are also invaluable to develop appropriate management plans (*Garrod, 2002*). *Davis et al. (1997)* assert that effective management planning for whale shark tourism needs both biological and recreational data. When complementing the ecological concern, recreational data and economic valuations can also be crucial tools to transparently determine appropriate management strategies such as visitation fees, licensing systems or other restrictions, as well as gaining public support on the implementation of such measures (*Ludwig, 2000*; *Catlin, Jones & Jones, 2012*; *Catlin et al., 2013*).

In this study, we improve current understanding of whale shark tourism by exploring the visitation patterns and economic effect of whale shark excursions in South Ari MPA in 2012 and 2013. To our knowledge this is the first study to model tourism metrics (expenditure, visitation and boat activity) in a MPA based on data collected with dedicated field surveys, rather than surveying a sample of the visitors which has been the traditional method of assessment. The results and recommendations we provide can be used to enhance the management of whale shark tourism at this location and encourage similar valuation studies in other wildlife attractions around the world.

## METHODS

### Study location

Officially designated a protected area in 2009, the South Ari MPA is the largest Marine Protected Area in the Maldives with a total area of 42 km$^2$. The legislative purpose of the

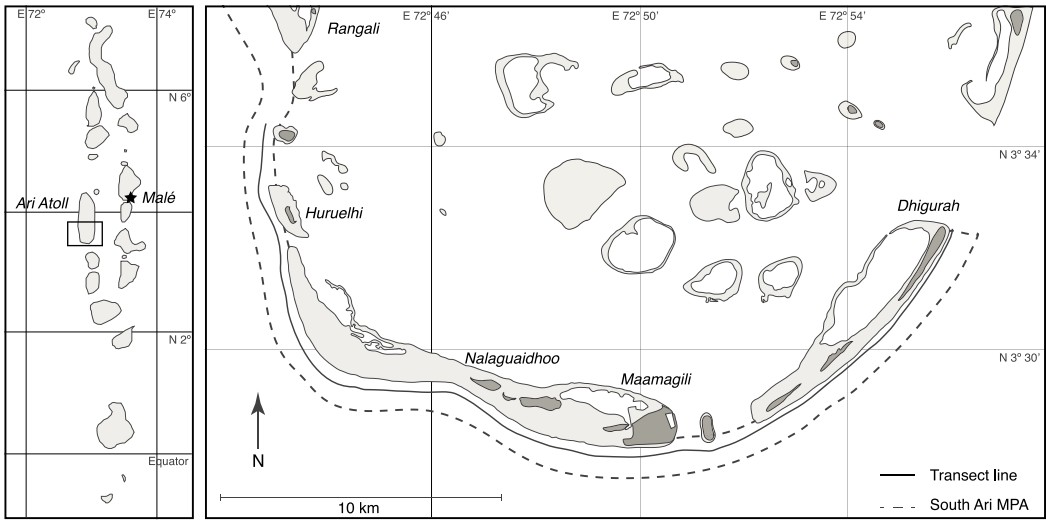

**Figure 1   Map of South Ari Atoll showing the South Ari MPA and the survey transect.**

MPA, according to the Maldives Environmental Protection Agency (2010), is to "protect and preserve a Maldivian aggregation of whale sharks, promote long-term conservation of the marine environment, and foster educational and scientific initiatives in the area".

The boundaries of the MPA extend along the seaward fringe of the South Ari Atoll from Rangali Island until Dhigurah Island, which encompasses 1 km of littoral zone measured from the reef crest (algal ridge) and includes the reef crest and 650 m–900 m of open sea (Fig. 1). The MPA boundaries represent the geographical area most commonly visited by tour operators for whale shark encounters.

## Whale shark tourism activity

Due to the geographical isolation of the Maldivian islands, tourists wishing to participate in a whale shark excursion in the South Ari MPA must go through a tour operator. Tour options are typically limited to dive centers, in-house operators at the resort the tourist is staying at, or with a liveaboard operator (locally called "diving safari"). Twenty-eight tourist resorts are located in the greater Ari Atoll, four of them on the MPA boundaries. Prices for whale shark excursions are varied and are exclusively determined by the individual operators. Guesthouses, situated in local islands as opposed to resort-exclusive islands, are a relatively new accommodation option. In this study we did not distinguish them from resorts or diving safaris due to their recent emersion and limited guest numbers.

## Data collection

From November 11, 2011, to December 31, 2012, the Maldives Whale Shark Research Programme (MWSRP) made 224 surveys along a 38 km linear transect section that coincides with the outer reef margin of the MPA. Surveys lasted $4.9 \pm 1.5$ h (mean $\pm$ SD) and were mostly started in the morning. Each vessel in the MPA within 500 m of MWSRP's

**Table 2 Number of survey days by year and weekday.**

| Year | Friday | Monday | Saturday | Sunday | Thursday | Tuesday | Wednesday | (all) |
|------|--------|--------|----------|--------|----------|---------|-----------|-------|
| **(a) During high season** | | | | | | | | |
| 2011 | 3 | 2 | 2 | 6 | 2 | 3 | 5 | 23 |
| 2012 | 4 | 14 | 5 | 11 | 13 | 11 | 14 | 72 |
| 2013 | 1 | 18 | 4 | 12 | 14 | 20 | 18 | 87 |
| (all) | 8 | 34 | 11 | 29 | 29 | 34 | 37 | 182 |
| **(b) During low season** | | | | | | | | |
| 2011 | 0 | 0 | 0 | 0 | 0 | 0 | 0 | 0 |
| 2012 | 1 | 3 | 4 | 0 | 1 | 0 | 2 | 11 |
| 2013 | 1 | 8 | 3 | 4 | 2 | 6 | 7 | 31 |
| (all) | 2 | 11 | 7 | 4 | 3 | 6 | 9 | 42 |

**Table 3 Boat types used in the study.**

| Type | Description |
|------|-------------|
| **Resort associated vessels** | |
| Excursion boat | 40–60 ft diesel engine traditional boats (dhoni) and 40–70 ft sailboats used for snorkeling excursions |
| Diving boat | 40–60 ft diesel engine dhonis adapted for one-day diving excursions |
| Sport fishing boat | 26–60 ft sport fishing boats and motor yachts whose primary purpose is recreational fishing by anglers |
| **Liveaboard associated vessels** | |
| Liveaboard | 70–140 ft boats that offer 10–30 guests to stay one or more nights at sea |
| Liveaboard diving vessel | 40–60 ft day boats for scuba diving and shore excursions from the main liveaboard |
| Tender | Outboard motor dinghies that support liveaboard operations |
| **Other** | Local fishing vessels, ferries and supply boats, PWC, military boats, dinghy sailboats, etc |

boat was documented by noting the vessel location, name, type and number of persons on-board. Surveys were part of MWSRP's monitoring program, which reduced operation intensity during tourist low seasons (March–September; Table 2).

During the surveys we estimated the location of the vessels with a handheld GPS unit. To determine the number of people on-board, a minimum of two observers individually counted the total persons on-board with the aid of binoculars. One person was added to the count if the skipper was not visible. The counts were repeated until there was consensus between the observers. The type of boat was selected between the options presented in the Table 3. All vessels not engaged in whale shark tourism were removed from the scope of this study.

We were only able to record spatial effort between October 2013 and December 2013, therefore the obtained boat distribution might only be an approximation. Although we were unable to survey the full extent of the MPA each day due to circumstances of external origin such as time, weather, or logistical constraints, we consider our surveys to be a representative approximation of a daily use census of the South Ari MPA as the same circumstances apply to tourists boats. This assumption does, however, imply that our expenditure and visitation figures might be underestimates of the actual values.

## Data analysis

We used an array of statistical models to estimate tourism metrics for the South Ari MPA for 2012 and 2013. We modeled six response variables: daily number of vessels associated to tour operators (resorts and liveaboards), daily number of visitors (from resorts, liveaboards and total number of guests), and daily direct economic expenditure on whale shark excursions.

We calculated the daily number of visitors by adding together the total number of persons observed on-board for each boat type. In order to control for the crew on-board, we subtracted two from the total number on-board each boat. Although occasionally there were more than two crewmembers per boat (especially on liveaboards), this imprecision is counteracted by the fact that in some cases we were not able to see and count all people on-board. To calculate daily direct expenditure we first multiplied the number of guests in a boat by the respective prices of a daily trip for each specific boat operator to determine the direct expenditure per boat. Subsequently all the expenditures per boat were summed. Because we surveyed the MPA only over a limited period of the day and because of the complications of counting the number of people on-board we consider our results to be conservative estimates of the actual tourism metrics.

Although it could change in the future due to the emergence of local community guesthouses and dive centers, for this analysis we included only resort and liveaboard associated vessels as currently they are the only ones considered to generate substantial whale shark tourism-based economic income. The prices per daily excursion were sourced through online queries based on boat name, type, and operator (if known). This search yielded the price of daily trips for 168 of the 568 vessels that frequented the MPA (Table 4). For the vessels that we were unable to obtain the 2013 trip price, a price average was allocated according to vessel type. Whale shark excursions are liable for a Goods & Service Tax under Maldivian law (Maldives Inland Revenue Authority, 2014); the associated taxes were not used in this study to determine the overall expenditure.

In the case of liveaboards, we estimated the daily price based on the total price of a trip per person in standard shared accommodation divided by the number of nights, without including taxes and service charge. We directly associated this expenditure with whale shark tourism in the South Ari Atoll MPA because the opportunity to encounter whale sharks is a primary reason for diving safaris to visit this area. Unlike the resort boats, we combined the boat types in the liveaboard category (liveaboard, diving vessel, tender) and assigned them a common price. We did this because it was usually possible to associate

**Table 4  Daily prices of a whale shark trip per person for each boat type (US$).**

| Boat type | Min. | Mean | SD | Max. |
|---|---|---|---|---|
| Liveaboard[a] | 90 | 247 | 68 | 381 |
| Resort diving boat | 17 | 102 | 61 | 200 |
| Resort excusion boat | 17 | 97 | 60 | 250 |
| Resort speed boat | 50 | 162 | 153 | 667 |

**Notes.**

[a] Liveaboards and associated vessels.

diving vessels and tenders to their respective liveaboards. Moreover, guests were often counted while on the support boats, not on the liveaboards.

In all six models we used the variables Season, Year and Day of the Week, mean daily Wind Speed (in order control for weather conditions), and the interactions between Day of the Week and Season, and Day of the Week and Year as explanatory variables. Roughly following *Shareef & McAleer (2007)* we considered that high tourist season occurs between October 1 and February 28 and low tourist season accounts for the rest of the year. Because there is no wind speed data measured from the MPA, we obtained daily means from the Blended Sea Surface Wind product from the National Climatic Data Center at the United States National Oceanic and Atmospheric Administration (*Zhang, Bates & Reynolds, 2006*).

To model expenditure, we fitted a linear model with generalized least squares (GLS) to the log transformed daily expenditure maximizing the log-likelihood. The GLS approach allowed us to account for heteroscedasticity, which improves the reliability of the coefficients calculated for the fixed effects (*Goldstein, 1986*). To select the most parsimonious model we used the Akaike information criterion (AIC), first determining the best weight and covariance structure, and then selecting the most appropriate fixed-effects set (*Zuur et al., 2009*).

To model the number of guests and boats for resorts and liveaboards—count data—we compared a generalized linear model (GLM) with a Poisson and one with a negative binomial error structure; the negative binomial distribution performed consistently better for all models (likelihood ratio test, $p < 0.001$). Although we detected a significant—albeit small—autocorrelation on the residuals of all models, we did not account for it. Instead, because of our priority on prediction precision (as opposed to coefficient estimation), we employed a multi-model inference approach that accounts for model inference uncertainty by averaging a set of candidate models (*Buckland, Burnham & Augustin, 1997*). Predictions were done with the AIC weighted average models that accounted for at least 95% of the evidence.

We used the models to daily predict the six response variables from January 1, 2012, to December 31, 2013, including those days when surveys were not conducted (due to limited sampling we did not predict any value for 2011). We then computed the annual number of visitors and annual expenditure by adding the daily results within each year. Because of the importance of quantifying the accuracy of our yearly estimates, we computed means and confidence intervals of the annual number of visitors by bootstrapping the models with

**Table 5 Yearly total expenditure and guests in the MPA calculated by adding daily model predictions within a year.** Confidence intervals (CI) and standard errors (SE) were calculated by jackknifing the expenditure model and by bootstrapping the guest models.

| Year | Expenditure (US$million) | | | Liveaboard guests (thousands) | | Resort guests (thousands) | | Total guests (thousands) | |
|---|---|---|---|---|---|---|---|---|---|
| | Total | SE | Bias | Total [95% CI] | Bias | Total [95% CI] | Bias | Total [95% CI] | Bias |
| 2012 | 7.62 | 2.69 | −0.70 | 26.27 [20.23, 37.06] | −2.09 | 45.07 [33.94, 55.57] | 5.76 | 72.37 [57.76, 85.43] | 0.52 |
| 2013 | 9.36 | 1.99 | 0.60 | 23.89 [18.43, 29.61] | −0.26 | 56.03 [46.35, 84.72] | 2.78 | 77.93 [65.55, 129.4] | −1.92 |

1,000 replications (*Young, Hinkley & Davison, 2003*). Due to the more complex parameterization of the expenditure model, we calculated the corresponding standard errors using the Jackknife method leaving one sample out at a time (*Efron & Tibshirani, 1986*).

All analyses were performed used R 3.0.2 with the packages nlme, glmulti, MASS, and bootstrap (*Calcagno & de Mazancourt, 2010*; *Canty & Ripley, 2013*; *Pinheiro et al., 2013*; *R Core Team, 2013*; *Venables & Ripley, 2002*).

## RESULTS

We estimated that mean direct expenditure on whale shark excursions was US$7.6 and $9.4 million in 2012 and 2013, respectively, with a mean total of 72,000–78,000 visitors per year for the same period (Table 5).

Daily direct expenditure on whale shark excursions ($E$) was calculated based on the most parsimonious model (Table S1):

$$log(E+1) \sim 1 + w + s + y + u \qquad var(\varepsilon_i) = \sigma_s^2 \times \sigma_w^2 \times \sigma_y^2 \qquad \varepsilon_t = \phi_1 \varepsilon_{t-1} + \eta_t \qquad (1)$$

where Day of the Week ($w$), Season ($s$), Year ($y$) and Wind Speed ($u$) are the fixed effects, and the variance of the residuals ($var(\varepsilon_i)$) is allowed to be different for each category of $w$, $s$ and $y$ ($\sigma_s^2, \sigma_w^2, \sigma_y^2$). The model also takes into account the temporal autocorrelation; the residuals at time $t$ ($\varepsilon_t$) are a function of the autoregressive parameter of first order ($\phi_1 = 0.123$), the residuals of the previous observation ($\varepsilon_{t-1}$) and noise. Detailed results of the model estimates can be found in Table S2. The daily number of guests and boats (both for liveaboards and resorts) were calculated from a weighted average of models that accounted for 95% of the evidence weight (Table S3). Predictions for the number of resort guests were based on all independent variables but not their interactions, whereas all other count models also included the interaction between Season and Day of the Week (detailed parameter estimates in Table S4a).

The effect of season was the largest in all models. While both liveaboard boats and resort boats visit the South Ari MPA in a given day more during high than low season, the difference between high and low season is three times larger for liveaboards than for resort vessels (Fig. 2D). There was a 60% decrease on the total number of guests, which was reflected on a 35% decrease on resort boats numbers and an 88% decrease on liveaboard

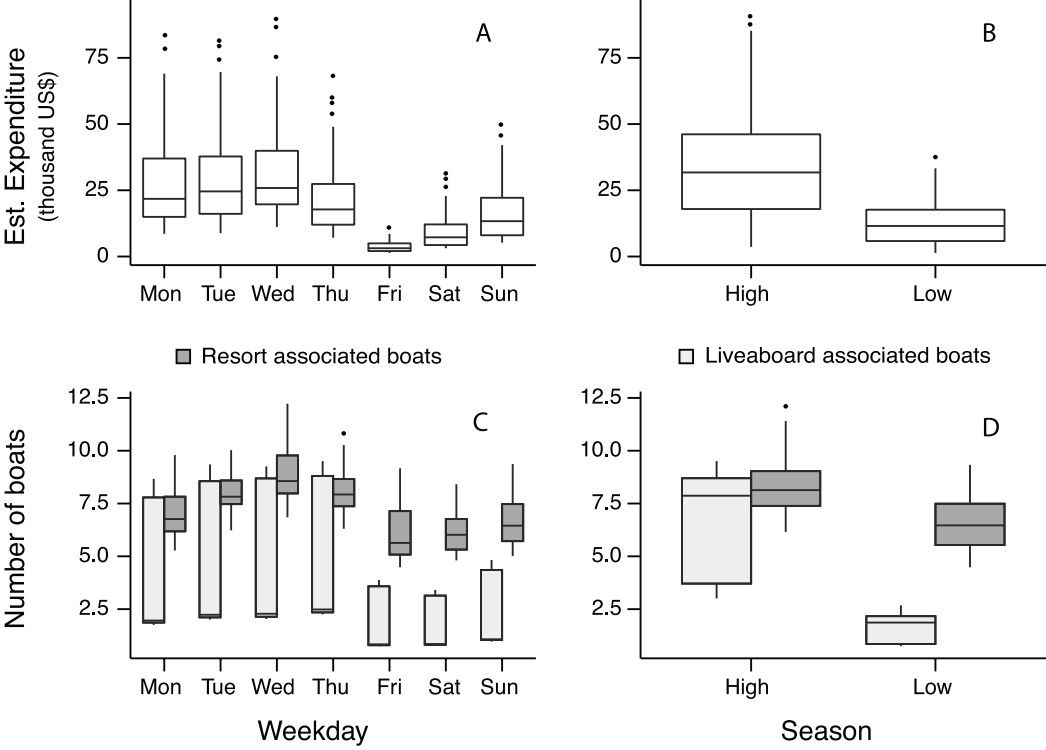

**Figure 2 Models results.** Values predicted by the expenditure (A and B) and the boat (C and D) models for different days of the week and seasons.

boat numbers, causing a 64% decrease in daily economic expenditure (estimates based on model coefficients; Tables S2, S4 and Fig. 2B).

Boat activity varied throughout the week—Wednesday being the busiest day and Friday the least (Fig. 2C). The vessel types encountered in the MPA also varied per weekday with liveaboard-associated boats being present much more from Monday to Tuesday than from Friday to Sunday. However, the presence of resort-associated boats was relatively constant during the week except on Wednesdays when there was a greater number of boats conducting whale shark excursions. In general, weekly patterns of vessel activity are similarly associated with visitors per day and expenditure per day (Fig. 2A). The estimated number of people engaging in whale shark tourism from resorts is not significantly different across the week, however, there are three times more guests from liveaboards on a Wednesday compared to Friday.

As expected, wind had a negative effect on the expenditure, for example a wind speed of one standard deviation above the average can cause a 13% decrease on the daily revenue. This negative effect is consistent in all models of number of guests and boats (Table S2 and Table S4).

Most of the boats visiting the MPA for whale shark tourism are encountered on a 5 km stretch between Nalaguraidhoo Island (Sun Island Resort & Spa) and Maamigili Island (Fig. 3).

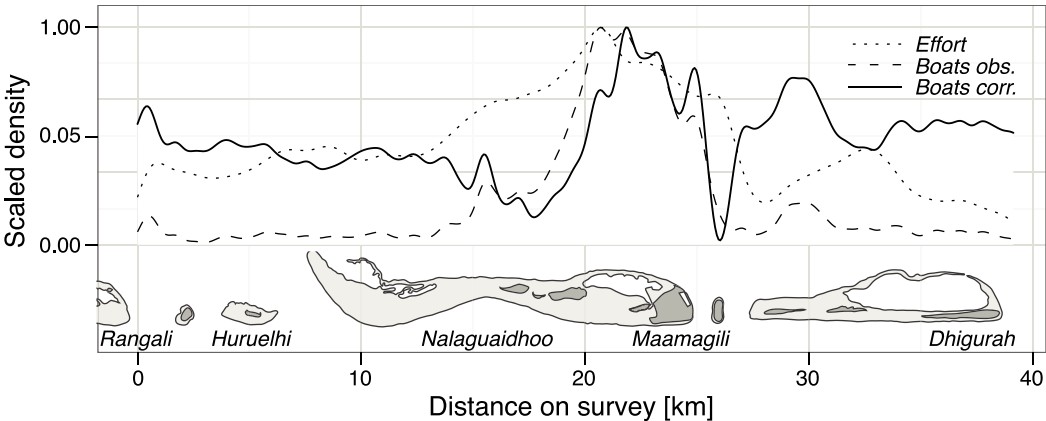

**Figure 3** **Tourist boat distribution in South Ari MPA.** Scaled density of survey effort in the South Ari MPA. We used a simple linear model to detrend the observed boat density and obtain a density corrected for effort (solid line).

## DISCUSSION

We estimate direct expenditure on whale shark excursions at US$7.6 ± 2.7 million (mean ± SE) in 2012 and $9.4 ± 2.0 million in 2013 based on an estimate of 72,000–78,000 tourists who are involved in whale shark excursions annually.

Both estimates were generated from the development of linear regression models as opposed to previous elasmobranch valuations estimates where expenditure surveys are administered to stakeholders and mean expenditure figures are multiplied by previously known guest numbers (*Catlin et al., 2010b*; *Anderson et al., 2011*; *Clua, Buray & Legendre, 2011*) . By taking into account temporal autocorrelation and using resampling techniques (bootstrapping and jackknifing) that allowed us to estimate uncertainty, we believe that our estimates can be statistically superior to valuations that select a sample of guests and average individual expenses, often without providing confidence intervals or any other measure of variability. Our method presents a novel, unified approach to calculate expenditure and visitation metrics in the absence of official tourist data, while at the same time it captures temporal variability that other methods are insensitive to.

For instance, despite the less frequent sampling during low season (which is reflected in a higher standard deviation for this stratum; Table S2), we detected, as expected, a clear significant difference on guest numbers and income generated by whale shark trips between seasons. This difference is stronger for liveaboards, which showed an 88% reduction in boat activity compared to a 35% reduction of resort boats. We also detected temporal variability on a weekly basis—Wednesdays bringing the most revenue and Fridays the least. Similarly, liveaboards visit the MPA significantly more from Monday to Thursday, probably due to weekly-based operations—Saturdays being the most common collection day of tourists in the capital city Malé (approx. 100 km away from South Ari MPA), while resorts show a nearly constant operation across the week.

Our estimate of $9.4 million for whale shark tourism in 2013 alone suggests that the value of shark tourism has experienced a marked increase over the last 20 years in the

Maldives, largely owing to a new focus on whale sharks. *Anderson & Ahmed (1993)* estimated that direct expenditure on shark diving tourism in the Maldives was US$2.3 million per year ($3.7 million in 2013, using U.S. Consumer Price Index). Our findings reinforce the observation that shark—especially whale shark—tourism has continued to expand over the last few years.

Similarly, other chondrichthyan species, such as Manta rays (*Manta alfredi*), are a major natural attraction for visitors to the Maldives. *Anderson et al. (2011)* estimated direct expenditure on manta ray diving and snorkeling excursions in the Maldives is around US$8.1 million ($8.7 million in 2013 dollars). Their estimates came from 91 dive sites throughout the archipelago with 157,000 visitors swimming each year, with a population of mantas in the order of thousands. Contrastingly, our $7.6–$9.4 million estimates of income from whale sharks come from just one site with 72,000–78,000 visitors per year and a population of 60–100 juvenile whale sharks (*Riley et al., 2010*). This underscores the significance of the South Ari MPA and the relatively concentrated industry while also highlighting the importance to implement sound management to ensure the sustainability of industry.

In the Maldives, with the emphasis on high-end resorts, the relative importance of diving has declined in recent years (*Anderson et al., 2011*). Although less developed, its whale shark tourism industry shows some similarities with the mature whale shark industry at Ningaloo Reef, Australia. *Catlin & Jones (2010)* explain that in Ningaloo the visitor profile has shifted from a specialist tourist interested in wildlife experiences to a generalist visitor with greater interest in the non-wildlife aspects. As whale shark tourism becomes more popular in South Ari, tour operators must put emphasis on a high-quality experience rather than in the encounter itself, especially in an industry where word of mouth is the key mechanism of promotion (*Catlin et al., 2010a*).

To increase the number of cases that meet and exceed guests' enjoyment and safety expectations and to minimize potential impacts of the industry on the whale sharks, stakeholders should promptly attempt to adopt management strategies. In fact, education, outreach, and regulative efforts can contribute to improved guest experiences (*Davis et al., 1997*; *den Haring, 2012*; *Techera & Klein, 2013*). Licensing of operators, which has been implemented in Ningaloo, has ensured minimal operation standards without it being perceived as an obstacle to business development. If licensing is flexible enough it can encourage continuous improvement of the operators (*Catlin, Jones & Jones, 2012*). An example to reduce crowding could be to focus resort operations on weekends since liveaboards visit the MPA more frequently from Monday to Thursday. Another example that comes from fisheries management, Individual Transferable Quotas, could limit the number of licensed boats in the MPA as a way to reduce crowding without dictating the actual number of people in the water with a shark at any time.

Alternatively, spotter planes can facilitate whale shark encounters by making searching more efficient and therefore dispersing operators among a greater number of sharks (*Rowat & Engelhardt, 2007*; *Catlin & Jones, 2010*). When the number of sharks available for encounters is limited, a code of conduct that encourages to "pass the shark"

from one operator to another after a mutually agreed time might improve guest experience and reduce potential impacts on whale sharks.

Because of the importance of up-to-date information in effective management we suggest the South Ari MPA stakeholders be directly involved in the collection of data on whale shark encounters and interactions. By supporting data collection using paper or electronic GPS based logbooks, the industry can obtain precise estimates, seasonal fluctuations as well as commercial feedback (*Department of Parks and Wildlife, 2013*). Stakeholder participation of this sort could be valuable to legitimize heightened management applications as well as assure timely stakeholder adoption of new regulations.

*Bhat, Bhatta & Shumais (2014)* found a large disparity between the economic value of atoll-based tourism in the Maldives and the amount of money that goes into environmental conservation. Collecting guest fees is now a well-established way to fund management strategies in protected areas (*Dharmaratne, Yee Sang & Walling, 2000*; *Thur, 2010*). It has been shown that as long as it is transparent, tourists are willing to contribute to the sustainable management of the whale shark experience (*Davis & Tisdell, 1998*). *Arthur (2011)*, in a willingness to pay survey, showed that tourists visiting the Maldives would be willing to pay an US\$106 ± 15 per trip (mean ± SD) to see sharks in their natural environment on top of the dive price and would donate US\$56 ± 6 towards a shark conservation fund. Exploring the guest willingness to pay is clearly an alternative that should be evaluated by stakeholders, managers and policymakers in the South Ari MPA if they are interested in improving or maintaining the quality of the ecosystem and the tourist experience (*Davis & Tisdell, 1996*; *Rudd & Tupper, 2002*).

Because of the scientific ambiguity and the many assumptions needed to value individual animals, we have refrained from ascribing a tourist value to the whale sharks in Maldives (*Catlin et al., 2013*). Our results, however, show that the Maldivian whale shark tourism industry is financially significant as it approaches 3% of the global shark ecotourism expenditure (*Cisneros-Montemayor et al., 2013*). Additionally, the results are indicative of the industry's local importance as a tourism driver that can generate revenue for local operators as well as the government. Based upon the expenditure rates for 2012 and 2013, the government would have collected approximately \$457,200 and \$748,800 (6% tax rate in 2012, and 8% in 2013), respectively, as a direct result of the whale shark tourism industry. This underscores the urgent need to manage this area to sustain the resident population of whale sharks by regulating use, so as not to exceed carrying capacity and limits of acceptable change (*Davis & Tisdell, 1995*).

Ecotourism projects are more likely to be successful when the target is a charismatic species and the management involves the local community (*Krüger, 2005*; *Gallagher & Hammerschlag, 2011*). Operators are in the best position to lead multidisciplinary and participatory processes to maximize tourist satisfaction while achieving protection goals and ensuring the long-term sustainability of whale shark encounters in the South Ari MPA (*Bentz, Dearden & Calado, 2013*). However, considerable discussion and deliberation will need to happen to determine the best approach that all stakeholders—including local communities, industry, and government—are willing to adopt to ensure a functioning

management system. This pursuit should be viewed as an iterative process with emphasis placed on evaluation and iteration based upon empirical findings.

## CONCLUSION

Based on empirical recreational data, we found that whale shark tourism in the South Ari MPA has been increasing in popularity and represents a significant wildlife tourism industry for the country, which follows the increasing popularity of the global shark tourism industry. Our findings are significant in that they bolster previous studies on Maldivian wildlife tourism that highlight the importance of the industry and urge for effective management. We think that this paper can contribute towards the establishment of an effective management system in the South Ari MPA and serve as a guide for other wildlife species and areas throughout the Maldives and elsewhere.

## ACKNOWLEDGEMENTS

We firstly thank the large team of MWSRP volunteers for the long hours on the dhoni collecting data and making this research possible. We thank Rachel Bott, Ben Fothergill, Katie Hindle, Rifaee Rasheed, Alissa Nagel, Michell NG, and the crew of Vilares I and other MWSRP and Conrad Maldives Rangali Island team members for field support. We also would like to thank Dr. Ameer Abdullah, Dr. Shiham Adam, Dr. Alistair Dove, Dr. Agnese Mancini, Morgan Riley, Dr. Chris Rohner, Dr. Brent Stewart and one anonymus reviewer for their insightful comments and suggestions, which greatly improved the quality of this manuscript.

### Funding

Funding for this research was made possible through the generous donations from MWSRP volunteers, sponsorship by Conrad Maldives Rangali Island, and programmatic and financial support from the IUCN Global Marine Program and Global Blue. The funders had no role in study design, data collection and analysis, decision to publish, or preparation of the manuscript.

### Grant Disclosures

The following grant information was disclosed by the authors:
Conrad Maldives Rangali Island.
IUCN Global Marine Program.
Global Blue.

### Competing Interests

Fernando Cagua, Neal Collins, James Hancock and Richard Rees are employees of the Maldives Whale Shark Research Programme; Neal Collins is an employee of the International Union for the Conservation of Nature. Fernando Cagua is a member of the MWSRP scientific advisory board.

## Author Contributions

- Edgar Fernando Cagua performed the experiments, analyzed the data, contributed reagents/materials/analysis tools, wrote the paper, prepared figures and/or tables, reviewed drafts of the paper.
- Neal Collins performed the experiments, analyzed the data, contributed reagents/materials/analysis tools, wrote the paper, reviewed drafts of the paper.
- James Hancock and Richard Rees conceived and designed the experiments, performed the experiments, contributed reagents/materials/analysis tools, reviewed drafts of the paper.

## Supplemental Information

Supplemental information for this article can be found online at http://dx.doi.org/10.7717/peerj.515#supplemental-information.

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
