# Peer review of "Whale shark economics: a valuation of wildlife tourism in South Ari Atoll, Maldives"

_PeerJ, doi:10.7717/peerj.515_

## Round 0.1 · original submission · Minor Revisions

· Academic Editor

Minor Revisions

The paper is well written, concise and needs some minor revisions to be acceptable for publication in PeerJ. Reviewer number 1 provides important guidance on the statistical treatment of the data that should be addressed prior to publication. In particular, assessing any potential effects of sampling effort bias and providing greater detail on the model outputs are essential in any revised manuscript. Reviewer 1 also identifies the need to explain some terminology to get the reader up to speed, and has a suggestion for a revised title which I think is a good idea.

Reviewer 1 ·

Basic reporting

Generally I found the paper well written. The authors seem to have been careful justifying their choices of models and data, and the story flowed well.

Experimental design

I have only a few major questions/concerns described below (by order of importance):
1. Sampling effort bias:
o In Fig. 3, could the curve observed be an artifact of sampling? Please include a similar curve (i.e., overall spatial coverage) for the sampling effort in the same chart.
o Also, it would be good to see the sampling effort in the temporal charts (weekly and per season).
o Authors did not seem to have accounted for this bias in the models – but it can highly influence results.

Validity of the findings

2. Prediction results and procedure used:
o It was not clear how the authors used the models to predict as there is not validation or prediction results shown (except for the one main result for each response variable). Is Table 5 a modelling result? Please clarify in the legend and present details on how you achieved these results.
o It seems that the model used the daily estimates obtained by the survey as response variable and then the authors predicted a response for each day of the year (?) and then added these all up to get to a year value? The procedure used needs to be clearly explained, and if the calculations were made per day, then these could be shown in a chart where you would also plot the sampled data for comparison.
o Because there were no surveys done in some months, I would suggest the authors to be careful presenting predictions for those months. Perhaps, show values with and without those months included.

Additional comments

Generally I found the paper well written. The authors seem to have been careful justifying their choices of models and data, and the story flowed well.
I have only a few major questions/concerns described below (by order of importance):
1. Sampling effort bias:
o In Fig. 3, could the curve observed be an artifact of sampling? Please include a similar curve (i.e., overall spatial coverage) for the sampling effort in the same chart.
o Also, it would be good to see the sampling effort in the temporal charts (weekly and per season).
o Authors did not seem to have accounted for this bias in the models – but it can highly influence results.
2. Prediction results and procedure used:
o It was not clear how the authors used the models to predict as there is not validation or prediction results shown (except for the one main result for each response variable). Is Table 5 a modelling result? Please clarify in the legend and present details on how you achieved these results.
o It seems that the model used the daily estimates obtained by the survey as response variable and then the authors predicted a response for each day of the year (?) and then added these all up to get to a year value? The procedure used needs to be clearly explained, and if the calculations were made per day, then these could be shown in a chart where you would also plot the sampled data for comparison.
o Because there were no surveys done in some months, I would suggest the authors to be careful presenting predictions for those months. Perhaps, show values with and without those months included.
3. All tables, equations and charts:
o Need better descriptions and need to include the meaning of all symbols used. E.g., eq. 1 is not explained in the text. Table S1, does not mention the meaning of symbols.
4. Economic analysis:
o A better introduction to the different types of economic analysis could be included in the introduction to make it clear that this approach is new. It starts directly with ‘direct spend’ without previous introduction to the subject. Also, I suggest using italics when referring to ‘direct spend’ to make it clear that the reference is to a method.
5. Title could be shortened and / or improved. Suggestion: ‘Economic impact of whale shark tourism in South Ari Atoll, Maldives’

I also include some minor comments:
- Abstract: change ‘2012-1013’ to ‘2012 and 2013’
- Replace ’72 to 798 thousand’ with 72 000 -78 000’ (throughout the paper)
- Include references: lines 13/14 and 36
- Line 73-75: I like this and think it could be developed in the first paragraph of Introduction (and then in the Discussion as well) to broaden the scope of the paper and its relevance
- Line 208-209: was this info used as strategy for sampling? How does it bias the results?
- Line 218: which techniques?
- Lines 249 – 253: not clear.

·

Basic reporting

No comments

Experimental design

No comments

Validity of the findings

No comments

Additional comments

This is a very well written article that addresses and important issue in the conservation management of marine megafauna ecotourism: how valuable is a living animal population to a nation's economy. Specifically, what direct value does whale shark ecotourism in Sth Ari Atoll, Maldives have? While I am not an economics researcher, the survey design seems appopriate and the statistical methods are similar to those used in scientific analyses. I can find no major flaws with the design, the analysis or the interpretation and have no substantive criticisms of the writing in this manuscript and so I recommend that it be published as is.

---

## Round 0.2 · accepted · Accept

· Academic Editor

Accept

The revisions on this article clearly address the minor concerns presented during review, and I think it is now ready for publication in PeerJ.